# Post-trauma coping in the context of significant adversity: a qualitative study of young people living in an urban township in South Africa

Rachel M Hiller,[1] Sarah L Halligan,[1] Mark Tomlinson,[2] Jackie Stewart,[2] Sarah Skeen,[2] Hope Christie[1]

## ABSTRACT

**Objective** Compared with knowledge of the post-trauma needs of young people living in developed countries, little is known about the needs of those in low-middle-income countries. Such information is crucial, particularly as young people in these environments can be at increased risk of experiencing trauma, coupled with less available resources for formal support. The aim of this study was to explore post-trauma coping and support-seeking of young people living in a high-adversity settlement in South Africa.

**Design** Semistructured qualitative interviews analysed using thematic analysis.

**Setting** An urban settlement ('township') in Cape Town, South Africa.

**Participants** 25 young people, aged 13–17 years, who had experienced trauma. Events included serious car accidents, hearing of a friend's violent death, and rape, and all reported having experienced multiple traumatic events. All participants identified as black South African and spoke Xhosa as their first language.

**Results** Social support was considered key to coping after trauma, although the focus of the support differed depending on the source. Parents would most commonly provide practical support, particularly around safety. Peers often provided an avenue to discuss the event and young person's emotional well-being more openly. Outside of social support another key theme was that there were numerous community-level barriers to participants receiving support following trauma. Many young people continued to be exposed to the perpetrator of the event, while there was also the realistic concern around future traumas and safety, community stigma and a perceived lack of justice.

**Conclusion** This study provides insight into how young people cope and seek support following trauma when they are living in a context of significant adversity and risk. Overall, most young people identified helpful sources of support and thought talking about the event was a useful strategy, but concerns around safety and trust could impede this process.

[1]Department of Psychology, University of Bath, Bath, UK
[2]Department of Psychology, Stellenbosch University, Stellenbosch, South Africa

**Correspondence to**
Dr Rachel M Hiller;
R.Hiller@bath.ac.uk

## INTRODUCTION

The experience of trauma can have a significant impact on a young person's developmental trajectory, including impacting

their psychological well-being and general functioning.[1] Evidence suggests that post-trauma support is one factor associated with a young person's mental health outcomes (particularly post-traumatic stress disorder (PTSD); for review, see ref.[2]). However, the vast majority of our knowledge on a young person's post-trauma support needs come from studies in high-income countries (HICs). Although the majority of the world's young people live in low-and-middle-income countries (LMICs), and high rates of trauma and adversity are often experienced in these contexts,[3] we know comparatively little about their post-trauma needs.

### Strengths and limitations of this study

► This qualitative 1:1 interview study explored the post-trauma support and coping of 25 Xhosa-speaking South African teens.
► All participants had experienced multiple traumas and lived in the Khayelitsha township, an urban settlement near Cape Town, South Africa.
► Due to the complexity of the environment, opportunity and snowballing sampling methods were used and we may not have captured young people with more severe or complex presentations.
► Transcripts were translated from Xhosa to English, and although steps were taken to ensure data integrity, some bias may have been introduced.
► This research captures a sample of under-researched individuals who had experienced a wide range of traumas (including intentional, accidental and hearing of, rather than directly experiencing, the event).

The post-trauma needs of young people living in LMICs, including where and how they may obtain support, are potentially vastly different to many young people living in HIC. For example, compared with young people in HICs, those in LMICs are more likely to live in environments that are objectively unsafe

and the concern of experiencing future traumas may be realistic, given evidence of substantially higher lifetime trauma exposures, both in terms of frequency and severity (eg, ref.[4]). Cultural differences and the availability of mental health resources may also lead to differences in support-seeking behaviour. For example, parents from HICs, interviewed about their child's post-trauma support, often report closely monitoring and responding to their child's emotional needs, including engaging in conversations about the trauma.[5 6] However, in LMICs parents may face a range of environmental factors that impact on their ability to provide emotional support to their child post-trauma, such as extreme poverty, significant community violence and a general lack of safety. Thus, the capacity to provide the kind of emotional support often reported in a high-income context may be quite different.

The aim of the current study is to develop an understanding of the support networks and coping strategies of young people who experience trauma within a context of significant adversity and risk. To do this, we investigated the coping and support needs of young people living in an urban settlement in South Africa, where much of the population live with high rates of adversity and poverty, and where rates of violence and trauma exposure are high. We sought to not only understand the ways in which these young people cope following trauma, but also identify key barriers to coping and support seeking, within this context. Such information is essential for informing culturally sensitive and feasible avenues for improving support, including by potentially building on already existing strategies or networks.

## METHOD
### Study setting
Ethical approval was granted by the University of Bath and Stellenbosch University Research Ethics Committees. Khayelitsha is an urban settlement ('township') near Cape Town in South Africa, with an estimated population of up to 1 million people. South Africa is considered an LMIC, where many people live with significant inequality and in extreme poverty. Khayelitsha is made up of both formal and informal housing, with many residents lacking access to electricity and sanitation. Alongside significant adversity, there are extremely high rates of violence and trauma, including sexual violence, gang violence and car accidents resulting in fatalities. Formal services to support children or adults following trauma are also lacking, while those services that do exist (eg, Rape Crisis Centre) are unable to cater for the large number of people who experience trauma within the community.

### Study sample and recruitment
Between March and October 2016, semistructured qualitative interviews were conducted with 25 Xhosa-speaking teenagers aged 13–17 years who lived in Khayelitsha. All interviews were conducted at the Khayelitsha Research Centre, located in Khayelitsha. To recruit participants, we used an opportunity sampling method. Community leaders (eg, Church leaders), local non-governmental organisations and community members were informed of the study and provided with information that they were able to distribute to parents or carers of potential participants. Research team contact details were also provided to the parent or carer of potentially eligible young people so they could contact the research team for further information. A snowballing sampling procedure[7] was also used to assist in the identification of potential participants, as study participants would often refer to other parents or carers of young people who had experienced trauma. Informed consent was provided by the parent/carer and informed assent was provided by the young person. All consent/assent forms were translated into Xhosa and were read aloud to participants. Participants and the parent/carer were provided with numerous opportunities to ask questions and were asked to summarise the study in their own words, to ensure clear understanding and allow any misunderstandings to be addressed prior to participation.

### Procedure
Interviewers were local data collectors who had experience in qualitative interviewing techniques and spoke Xhosa as their first language. Prior to beginning the interviews, all interviewers received further training in qualitative techniques and interviewing children. Training was provided by an experienced qualitative researcher who had extensive experience working with young people who had experienced trauma or maltreatment and had experience in trauma research (JS). Training included sessions on the nature, purpose and conduct of qualitative research, the possible impact of trauma on individuals, in-depth discussions regarding the aim of each interview question, and mock interviews and feedback from the research team. Due to the sensitive nature of the interviews, it was not appropriate to have the authors present during the interview to monitor interview quality. However, interview quality was continually monitored during the study. Interviewers kept their own reflective logs throughout the study and detailed feedback on interview content and process was provided through regular supervision.

Female participants were all interviewed by a female interviewer. Male participants were primarily interviewed by a male interviewer, with two exceptions where it was not practically possible. The interviewer was not known to the young person. While there was the option to have their carer present during the interview, all interviews were conducted 1:1 with the adolescent and interviewer. Interviews were audio taped and were then translated and transcribed verbatim into English by an experienced bilingual translator, who had not been involved in the interviews. Transcripts were then reviewed by the interviewer, including to add any additional non-verbal information. Any disagreement was resolved through a consensus meeting that included referring back to the audio recordings.

As a thank you for volunteering their time, participants were given a 120ZAR (approximately US$8.23) voucher, as per standard policy of the research centre. As per standard risk management procedures, where appropriate, referrals were provided to local mental health services for either the young person or parent/carer.

## MEASURES
### Semistructured interview

The semistructured interview was developed based on the literature on qualitative interviewing techniques (eg, ref.[8]) and previous qualitative work on post-trauma experiences (eg, refs.[6 9]). The interview guide covered four broad areas: (i) basic descriptive information, (ii) the child's views on what they found help or unhelpful following their experience, (iii) ways they coped (or didn't) post-trauma and (iv) their perspective on ideal support. Questions were largely open ended (eg, 'Was there anything you did to try and make yourself feel better?', 'Was there anything that happened that made it hard to cope?, 'Do you think anyone could tell how you were feeling?', 'If you had a friend go through something similar how might you help them?', with some more direct prompts used to seek clarity (eg, 'Was that helpful?'). Interviews lasted 45 min to 1.5 hours.

### Trauma-related descriptive information

To obtain some basic trauma-related descriptive information, all young people completed the traumatic events checklist from the University of California, Los Angeles PTSD Reaction Index[10] and the PTSD symptom scale of the Child PTSD Checklist.[11] The latter was chosen as it is a measure of post-trauma distress that has been validated in non-Western populations, including South Africa.[12 13] The Child PTSD Checklist is a validated 28-item self-report questionnaire that assesses PTSD symptoms on a four-point scale (not at all; sometimes; most of the time; almost always), based on the 4th edition of the diagnostic and statistical manual of mental disorders (DSM-IV-TR). The symptom checklist was completed in relation to the event that the teen reported finding most distressing now. A likely PTSD diagnosis is considered where the young person reports 'most of the time' or 'almost always' to experiencing at least one re-experiencing symptom, three avoidance symptoms and two hyperarousal symptoms, consistent with a DSM-IV-TR diagnosis.[14] A partial diagnosis of PTSD was given where the young person endorsed at least one symptom across all three symptom domains.

### Data analysis

Using NVivo software, the translated interview transcripts were analysed by RH using thematic analysis to identify main themes and patterns in the data.[15] This approach was chosen as it does not rely on a specific theoretical framework and enables detailed exploration of the data.[15] To begin with, RH read all transcripts to gain an understanding of the overall data (immersion[15]). Next, each transcript was systematically coded. Saturation was achieved at transcript 17 (ie, no new themes were generated). Following coding, the first five transcripts were rechecked, to ensure coding had remained consistent. Codes were then grouped to form overarching themes. To determine reliability, HC then read all (uncoded) transcripts and generated her own list of key themes, blind to the themes derived by RH. Themes were discussed at a consensus meeting between RH and HC, with strong agreement between the two coders. Finally, key themes were discussed with the Khayelitsha research team (ie, peer debriefing), to assess credibility and ensure understanding of potentially unique cultural details.

## RESULTS
### Descriptive information

The sample comprised 11 girls and 14 boys, mean age 15.1 years (SD=1.4). Almost half of the sample (n=12) lived with both biological parents, five young people lived with their mother only, three with a biological parent and a step-parent, three did not live with either biological parent (eg, lived with neighbour, other relative) and one lived with their father only. One young person did not report with whom they were currently living.

Participants reported exposure to between 2 and 10 events on the UCLA traumatic events checklist (M=5.2 events, SD=2.4). From this checklist, of the 25 young people interviewed, the most commonly reported trauma was witnessing someone being beaten, shot or killed (80%), followed by hearing of the violent death of a loved one (68%), the young person themselves being beaten, shot at or badly hurt in the community (60%), seeing a dead body in the community (60%), being beaten at home (40%), seeing a family member beaten (40%), being around war (36%; likely referring to gang violence), sexual assault (36%), natural disaster or fire (32%), having a very painful or scary medical procedure (32%) and being in a bad accident (28%). Descriptive information about the trauma identified as most significant by the young person is presented in a supplementary table. Four young people (16% of sample) met criteria for a likely PTSD diagnosis, based on the self-report measure. A further 10 young people (40% of sample) met the criteria for partial PTSD. Only two participants reported receiving formal counselling, both following rape.

## THEMES
### Theme 1: social support was central to coping

When asked what they found helpful following their experience, almost all participants discussed the role of someone in their social circle who had provided them with helpful support. There were also some cases where the wider community (ie, outside their direct social circle) also contributed to providing a sense of comfort or security following the trauma. The primary sources of support

identified by the young people were (i) their parents or a close family member, (ii) their peers or teachers and (iii) the wider community (particularly around Church). The type of support provided by each source was qualitatively different.

## Subtheme 1: support from a parent or carer was important

The majority of participants thought their parent(s) or relatives provided good support, particularly female carers (eg, mothers, grandmothers). Indeed, this was the most commonly discussed source of support. While support from a parent or carer was considered important, this support was rarely focused on discussions around emotional well-being. Overall, most thought their parent or carer could tell that something was wrong with them (ie, that they had changed), but would also report that they were rarely asked about their thoughts or feelings about the event. Conversation was often around the facts of what happened, so the parent could understand changes in the child, rather than around the young person's thoughts, feelings or reactions to the event. Primarily, participants viewed the parent's role as providing them with a sense of security or safety that may prevent future harm. Open discussions around the young person's emotional well-being following the trauma occurred in the minority of cases, with parent or carer support generally focused on encouraging forgetting and trying to ensure practical safety. The encouragement of forgetting often involved removing reminders of the event or specifically directing the young person to not think about, or discuss, the event.

Participant P: Maybe when I was thinking about that thing, she [mother] will come to me … she changes the topic and I would find myself laughing. We will then move on talking about something else until I forget about that. We will talk about things and buy drinks while chatting until I found myself forgetting about that.

Participant A: … she [mother] would ask me to stop thinking about this thing or ask me to go to my friend or ask me to sleep.

Participant H: For an example when we are watching the movie with the gunshots she [mother] would change the channel, not wanting me to listen to the gunshots.

In many cases, to help the child to 'forget' and to help ensure their safety, the parent would change the child's routine or send the child to live in another area, so they were away from reminders of the event. Participants generally regarded this as positive support, allowing them to feel more secure.

Participant A: I went away to my aunt's place just for two weeks for me not to see them, thinking that if I don't see them I will be fine …

Participant H: She does not want me to stay with friends most of the times. She wants me to be in the house about 17h30 because she tells me that it is

dangerous in the street … She wants me to be in the house most of the time.

Of note, in cases of sexual assault or rape, mothers were seen to provide a particularly crucial role in pursuing formal criminal justice. Often as soon as the mother found out about the rape they would take their daughter to the hospital for tests and then straight to the police station. Despite the oft-reported failings of the criminal justice system, having a parent pursue some justice or file a case was seen as an important step to coping.

Participant R: My mother took it seriously and did everything … Like she immediately took me to the police and we slept at the hospital … and the statement was taken there and then and I said everything.

While most participants reported that the parent/carer was a positive source of support, there were a minority of cases where the participant reported that they did not find parental support helpful, particularly from fathers. This was most commonly reported by males, where the event had involved gang violence. It was also reported across gender and trauma type, where there was ongoing domestic violence or drug or alcohol abuse in the home, or the young person thought the parent would act aggressively or be angry at them (eg, beat them, shout at them).

Participant Q: He shouts at me and tells me that why did you do this … My father was really upset by this [event], I would see him when he looks at me that he hates me, he would look at me like he sees, I don't even know what, he was disgusted with me.

## Subtheme 2: peers were important for emotional support

Talking to peers was a subtheme related to the importance of social support. Many participants reported that their schooling was affected by their experience, including failing tests or whole grades, as well as their general behaviour (eg, becoming socially withdrawn). Thus, school was often discussed as a place where someone would notice that the young person had been through a traumatic experience. Peers then became particularly important as a source of emotional support. While there were two cases where participants reported being ostracised by their peers (both where the event had been rape), overall, compared with parent support, support from peers was often more likely to focus on talking about the event and emotions. The young person trusting the friend (eg, that they would not gossip to others or judge them) and shared traumatic experiences (ie, that their peer would have a good understanding of the experience due to their own similar experience) were key to this peer support. In many cases, the young person discussed a kind of informal self-help group, which was often also actively encouraged by teachers.

Participant A: I felt it helpful because after sharing with them [friends] I stopped feeling lonely like feeling alone as if there's no people, at least there

were people who are in my situation you see at least they feel what I feel you see.

Participant L: There is someone I told after a week, it was my coach because he asked me why I was performing like that in when playing ball and I was hurt and I told him about the incident that happened and he understood.

Participant S: … the teacher told me that in that school they have groups, whereby people talk about their problems and not be shy about them so I also told them my problem and the children liked me.

Participant X: They understand my situation at home [domestic violence] and I also understand theirs at home.

### Subtheme 3: wider community support could also be helpful, particularly in relation to religion

Finally, outside of the young person's more immediate social support, many reported that the wider community could be a positive source of support, particularly concerning religion or traditional healing. Christianity and traditional healing play a key role in the Khayelitsha community. Indeed, 'God's will' or witchcraft were considered key explanations for why the event occurred, including why someone had caused the event or why the young person had managed to survive the event.

Participant F: I can say that God is someone that saves people. I do not know how He saved me but I will say, I was saved by him.

Participant L: I think the reason is that there are people that are after me, because I am a child who is all right, so they take me as if I think that I am above them because there is money at home, so I think it's the witches which are in the village I come from.

Following trauma, many young people, along with their parents, obtained a sense of comfort or support from the Church community, including finding supportive people through Church or a general sense of comfort and protection from prayer.

Participant N: I told my best friend first at school the one who goes to church at school and said that my friend I am not feeling good about myself … Even at home I am seemed to be the bad person to the parents in the location. I wish I can change. I told my friend and my friend said that we must go to church to my mother. Let's go to the church you will be relieved.

Participant P: I sometimes think about it. I tell my mom that this is coming up and my mother will ask us to pray. She will pray and pray and I would sleep.

### Theme 2: 'forgetting' in considered key to coping after trauma

The majority of participants focused on the importance of 'forgetting' as the ultimate way of coping following trauma. When discussing how they coped, any social

support they received, their views on ideal support or how they might support a friend who had a similar experience, much discussion centred around how to 'forget' the event. In some cases, strategies to 'forget' were quite adaptive, while in other cases strategies were considered maladaptive (eg, drug use). As previously discussed, the importance of 'forgetting' was often also encouraged by their social support systems and particularly parents.

### Subtheme 1: adaptive strategies to 'forget'

In some cases, strategies used to 'forget' appeared relatively adaptive. Despite the focus on the importance of forgetting, in many cases the discussion suggested the young person was actively processing their experience as a means of moving forward. Indeed, the majority of participants endorsed the idea that talking to someone about the experience was key to coping and overcoming the trauma.

Participant L: [the] more you talk, the more you forget, the more you are being comforted by someone, the more you feel good.

Participant V: I felt alright because I also met with the doctor and spoke to someone … and I felt free … if you speak about the thing [event] you forget about it.

Beyond talking about their experience, it was also common that young people would engage in play or fun activities with their friends, as a means of 'forgetting', at least in the short term.

Participant C: I went to play because when I sit at home I get lonely and this thing [event] come so I need to play and play all over again.

Participant X: I was really playing [games] because I wanted to forget, I would really play.

### Subtheme 2: maladaptive strategies to 'forget'

In contrast to the more adaptive coping strategies, another strategy to 'forget', largely reported by boys, was the use of drugs and alcohol. While in hindsight it was mostly acknowledged that this was not an adaptive coping strategy, for many of the males it was used as a way of 'forgetting' their experiences, at least in the short term.

Participant K: There were a lot, there were a lot of things I did that year … that's when I started being wild, for me to forget I would think of smoking, that is the year I started to smoke dagga [drug], before that thing happened I would think of smoking only on certain events but after that thing I wished I could not go home because I would get there and everyone would be feeling down because of that thing. I wish I could just sit with the guys and maybe drink … When I look back now, they [drug use] didn't help me but when those things were happening I thought they were helping me because when I got home I would eat and sleep then wake up in the morning and go to school. I would eat and sleep because I get hungry

when I have smoked, I would eat and sleep and that made me happy.

### Theme 3: community-level and family-level barriers to coping are present

A final key overarching theme was the significant ongoing barriers all participants faced when trying to cope with any single traumatic event. Here, four subthemes were identified as key barriers to coping: (i) lack of safety/risk of future trauma; (ii) continued exposure to the perpetrator; (iii) a lack of justice, either through criminal channels or vigilante justice; and (iv) concern around community gossip or stigma. These themes were considered barriers as the young person either reported that it had prevented them from disclosing to people about their experience or from 'forgetting' (something many young people identified as important), or where it was a source of significant ongoing anger (eg, a point that they felt 'stuck' on, preventing them from moving forward), or where the young person perceived it effected their sense of safety or security after the event.

### Subtheme 1: general lack of safety

All of these young people had experienced multiple, serious traumatic events and were managing their post-trauma psychological adjustment within an extremely complex environment. A general lack of physical safety was identified as a key barrier to overcoming any single trauma.

Participant D: The problem is that I do not see any safety.

Participant E: … I know that everything can happen anywhere, there is nowhere that something cannot happen, even at home it can happen. There is [no one] that can say it may not happen, anything can happen.

A lack of safety and instability was particularly reflected where the event had stemmed from vigilante/mob justice, perpetrated by community members, often over small-scale crime such as stealing a small amount of money or food.

Participant O: When I heard this thing [friend had been murdered by vigilante mob] I thought to myself, if there is someone who can kill for an amount of money such as R400 [~£20] and second-hand sneakers and biscuits. How is this thing? … I lost my mind … I was someone who was always scared [after event] because I didn't like to walk where he was beaten up because they might say I am his friend and also want to do something like what he did.

While in many cases this general lack of safety reflected the unsafe community outside of their home, there was a minority of cases where the ongoing unsafety and instability was a result of continued exposure to trauma and stress within the home. This included domestic violence

and drug and alcohol abuse within the home, while some shared concerns around homelessness, or managing chronic illnesses such as HIV or tuberculosis. In such cases, coping following any single event was largely overshadowed by day-to-day coping with other significant ongoing traumas or stressors, as demonstrated below.

Participant S [discussing ongoing events]: They would go to the street committee and the street committee would come and talk to him [father] and he would tell them that he will never do it again but he would do it again and the street committee has now stopped coming because they say that they are tired of this thing … When I am being beaten, there is no one that comes and tries to stop it. They just leave him to beat me until he feels like stopping.

### Subtheme 2: continued exposure to the perpetrator

The majority of young people interviewed continued to be exposed to the perpetrator of the trauma, making it difficult for them to move forward or 'forget' about the event. This barrier was often associated with engaging in avoidant coping strategies and changes in routines in attempts to maintain safety.

Participant D: I think at home with my parents we can go stay elsewhere [move] because I can't stand facing these people that did this.

Participant W: I also know his brother's car, the one they came with. I would think I must look out for that car on the road so that it does not just come and take me.

Continued exposure to the perpetrator was particularly highlighted as problematic by female participants who had experienced sexual assault.

Participant B: We are from the same village but when I see him [perpetrator] I run home or stay in my house.

Participant R: Maybe he would eventually arrive … and maybe kill me because that is what my mother said, she said that what is next is dying, when they say that it was planned … what is next is death, they will make sure that they kill me.

Participant V: He [perpetrator] passes by the street, he lives next to my home.

### Subtheme 3: perceived lack of justice

Another key barrier to coping post-trauma was the young person's perception of the lack of justice. In some cases, this was a failure to achieve justice through official police channels. However, in many cases this referred to the young person perceiving that there was also a lack of vigilante justice. While a lack of justice was often linked to continued exposure to the perpetrator, it could also be related to the reaction of the perpetrator, their family or friends. When asked what was helpful, many young people stated that justice was central to whether they felt

they could 'forget' or move on from the event. In many cases, they felt that the system was unjust, leaving them feeling angry and unsafe.

Some participants referred to the widely held cultural belief that the perpetrator's family should care for the victim. This may include visiting them in hospital and paying for damage or bills or paying some compensation to the victim's family. In cases where this had not occurred, young people said that they struggled with this.

Participant K: That [perpetrator's] family did not care about me. After I was shot they did not care about me, they just left everything. My family said some harsh words and said that there is something that will happen there, the police will make sure that they are arrested because they just left everything. They only went once in hospital and they never went again, they just left everything on my family.

Participant J: I am thinking that person did it on purpose because if he hadn't done it on purpose, he would have gone home apologised or pay so that the house is fixed, I am thinking that person did it on purpose.

While many young people expressed a desire for the perpetrator to be arrested, to allow them to 'forget', in the absence of formal criminal justice, many expressed a desire for revenge or vigilante justice to occur.

Participant B: I wanted him to be beaten up to death because maybe he is going to continue with what he did to me, rape other girls, because if they/he is arrested today then you will find them released from jail again and still continue with what they are doing.

Participant J: I felt that I wish that person would crash with his car and die and that would be the end of him or for him to be found and made to pay for the house [damage] or for him to be beaten up—things like that—or for him to be arrested.

### Subtheme 4: community stigma or gossip

While in the case of religion the wider community could provide a sense of comfort and safety (as previously discussed), many participants also discussed their concerns around community gossip or stigma. This was considered a barrier as participants identified it as a key reason for why they would not discuss the event with friends or community members, despite recognising that this was important for coping. In some cases, this subtheme also arose because the young person was afraid that people would come up to them and ask them to talk about it when they did not want to. For boys in particular, the concern was often about getting a reputation as a gang member.

Interviewer: Why did you not tell your friends about what happened?

Participant C: Because if I told them they will talk about it everywhere, and people will come to me and ask what happened.

Interviewer: Why didn't you tell the whole details of the story?

Participant G: Because people like to go around gossiping and that will make [victim] to be uncomfortable.

Participant I: … people from [residential area] gossip, when you pass there, grandmothers are talking about that thing [event]. When there is a lot of them standing, it gets full and they talk about that thing, everyone knows about it, they shout it out loudly and it even gets heard by kids who were not there and they ask us what happened and we tell them we don't know.

For girls who had experienced rape or sexual assault, the concern was often around community members labelling them a liar or accusing them of being promiscuous or damaged. Stigma and gossip was particularly problematic in cases where the young person was still exposed to the perpetrator or family members of the perpetrator.

Interviewer: Why don't you walk with your friends anymore?

Participant W: Because they now gossip about me when I walk with them and talk bad [about me], it is better that they gossip while I am not there.

### DISCUSSION

Compared with knowledge on post-trauma coping in HICs, far less is known about how young people cope and seek support following trauma in LMICs, where the post-trauma environment can be impacted by significant adversity and risk. To address this, the current study explored the coping and support needs of a group of young people living in a township in South Africa who had experienced trauma. Most of the participants could identify helpful ways to cope or helpful support networks post trauma, particularly highlighting support from parents or friends, and the role of religion. However, all described significant barriers to coping, many of which stemmed from a lack of safety and a perceived lack of justice within the community.

With the exception of a few cases, young people generally reported that caregivers, particularly female caregivers such as mothers or grandmothers, were a useful source of support. This support was largely around providing the young person with a sense of safety, rather than encouraging any direct discussion of emotional well-being. Indeed, young people rarely reported that a carer or parent approached them about their feelings following the trauma. While results are in contrast to studies in some Western HICs, where parents would commonly encourage discussion about the event and child's emotions,[5 6] findings are consistent with a recent

qualitative study that explored parent perspectives of supporting children post trauma, within this same community.[9] Here, while parents often saw changes in their child's behaviour following trauma, the key themes to emerge were ensuring the child's practical safety and the encouragement of avoidant coping, including forgetting.[9] Parents reported that it would be particularly unusual to ask their child about their feelings.[9] In line with parent reports, current findings suggest that there is an avoidance of discussion around emotions in the family home, although many young people reported that they would have liked the opportunity to discuss their feelings with their carer. Overall, participants often described a degree of dissonance between wanting to avoid reminders and forget the event, but also wanting an opportunity to discuss their feelings and process their experience(s).

It is essential to consider the high-risk context when interpreting findings in relation to coping and caregiver support. All young people in the sample reported having experienced multiple traumas and many continued to be exposed to the perpetrator of the index trauma. Faced with this genuine lack of security and with explicit reminders of the experience, it is understandable that avoidance may become a key coping strategy both from the perspective of the young person (ie, avoiding going outside) and their parent (ie, keeping them inside more, avoiding conversations about the event). While avoidant coping is considered a risk factor for poor psychological outcomes following trauma in HICs,[2] it may be adaptive in a context where safety is compromised in an ongoing way. Preliminary research provides some support here, with research on high-risk communities living in HICs finding that avoidant coping can be associated with better psychological and functional outcomes for youth.[16 17] That is, in high-stress environments engaging in avoidant coping (ie, avoiding reminders of the event, suppressing thoughts of the event) can actually be adaptive, at least in the short term; longer-term implications of this coping style remain unknown. The potential utility and consequences of avoidant coping in high-risk LMIC contexts remains an important area of further investigation, as research seeks to understand the balance between physical safety and emotional well-being in such complex environments. Given that avoidant coping is targeted in most trauma-focused psychological interventions, comparisons between the short-term and long-term consequences of behavioural and cognitive avoidance in this context remains crucial.

Beyond support from carers, many young people turned to peers as a means of emotional support. Peer support has been identified as important for young people's post-trauma coping in HICs following accidental injury[18] and interpersonal trauma,[19] as well as in research with adolescent refugees[20] and former child soldiers.[21] Indeed, in some cases, emotional support from peers can be considered more important for coping than support from parents or formal support services.[20] As in

the current study, qualitative research in other high-risk groups has found that concern about stigma meant young people were selective about who they would disclose their experiences to, with trusted friends often identified as key sources of support. Linked to trust and understanding, many young people in the current study reported that they sought emotional support from peers who had been through similar traumatic experiences. The particularly supportive role and trustworthiness of peers with shared experiences has also been found in work with teens who had experienced sexual abuse in Switzerland.[19] While in this HIC study this was discussed by a minority of participants, it was a theme discussed by the majority in the current study. In this high-risk context, the majority of young people knew friends who had been through similar traumas, with whom they could openly talk about the event and seek advice for coping. In some cases, this involved providing space to discuss the event and emotions, while in others it may simply involve playing games or sports that allow the young person to feel a sense of normalcy, at least in the short term. In some cases, the school or teacher had begun actively encouraging the creation of peer self-help groups as a means of providing support for the high number of traumatised youth.

Findings suggest that school and peers may be a key avenue for providing support to young people post trauma. There is already some evidence of the effectiveness of school-based interventions for PTSD in developing countries (eg, ref.[22]), as part of the recommended stepped-care or scaled model.[23] School-based programmes that include some parental involvement may be particularly beneficial at promoting parent–adolescent conversation around potentially sensitive topics.[24] Parental involvement could be especially relevant in the current group, given the previously discussed mismatch between parent beliefs about coping (ie, that they would not usually talk to their child about their feelings[9]) and young person beliefs (ie, that they would like the opportunity to discuss the experience). A particular benefit to building capacity for emotional support within a school context is that it would allow the support to get directly to the young person, rather than relying on the young person to seek out formal help; given both the limited access to formal psychological support, along with anecdotal and empirical evidence that young people in these high-risk contexts can be unlikely to engage with formal services (eg, ref.[20]).

Results of this study should be interpreted in the context of some key limitations. First, due to the complexity of the environment, an opportunity sampling method was used. Thus, despite half of the sample meeting criteria for PTSD or partial PTSD, even years following the event, our sample may still not capture the most severely psychologically impacted young people within this community. Indeed, concern around community stigma and not wanting to talk about the event(s) was identified by the research team as a likely barrier for recruitment. Due to the recruitment method, the study would also not have

captured young people where the parent was unaware of their trauma experience. Second, due to sensitivity and concern around the appropriateness of collecting demographic information from young people, we were unable to collect data on more detailed demographics such as the type of housing (eg, formal vs shack) or access to electricity or sanitation. However, all were living in a settlement where the community largely lives with significant poverty. Finally, although feedback on themes was sought from the local research team to ensure findings reflected the sociocultural context, we were unable to conduct respondent validation with the young people. Despite these limitations, this research captures a group of under-researched individuals who had experienced a wide range of traumas (including intentional, accidental and hearing of, rather than directly experiencing, the event) and lived in a range of settings, ranging from young people who lived with both biological parents to young people who had been orphaned. Across this range consistent themes continued to emerge.

Understanding the complexities of the environment faced by young people in LMICs, such as what is captured by this study, is essential information before exploring key questions through quantitative means or designing interventions, where validity, feasibility and appropriateness are all important considerations. There is a clear need for future research on young people's post-trauma environment in the context of significant environmental adversity and risk, both in LMICs and for high-risk groups in HICs. Within LMIC contexts, further qualitative and quantitative work will be important for understanding the views of a variety of stakeholders, including carers, teachers and social workers, as well as determining whether particular coping styles may facilitate better or worse psychological adjustment, and how any intervention could be designed to make use of established support networks. While the current study highlighted the complexities of balancing physical safety and emotional well-being in high-risk communities, it also showed that many young people are actively seeking emotional support from their peers and that school may be a useful avenue for building capacity around supporting the post-trauma emotional well-being of these young people.

**Acknowledgements** Thank you to everyone at the Prevention Research for Community Family and Child Health Centre in Khayelitsha, for making this research possible, and in particular Vuyolwethu Notholi, Phumza Gqwaka and Mzi Sigamele for running the interviews. Thank you also to the participants who were willing to share their stories.

**Contributors** RMH is the lead researcher on this project and the PI on the British Academy Small Grant. She led the conceptualisation of the project, project management, data analysis and the write-up of the paper. SLH was involved in the conceptualisation and design of the project, including being a co-I on the British Academy Small Grant and PI on the travel grant. She was particularly involved in the design of the semistructured interview. She contributed to discussions about themes and provided feedback on the manuscript. MT was involved in the conceptualisation and design of the project, including being a co-I on the British Academy Small Grant. He is the director of the Prevention Research for Community Family and Child Health Centre in Khayelitsha, where this study was conducted. He also provided feedback and edits to the manuscript. JS is the co-director of the Research for Community Family and Child Health Centre in Khayelitsha.

She contributed to discussion about the design of the study and particularly the semi-structured interview. She contributed significantly to project management throughout the study, including managing the team of research assistants and providing training on qualitative interviewing techniques. She also provided feedback on the manuscript. SS contributed to project management during recruitment, including monitoring and providing feedback to interviewers and research assistants throughout the recruitment process. She provided feedback and edits on the manuscript. HC was involved in the thematic analysis of the data and contributed to the write-up of the manuscript.

**Funding** This research was supported by a British Academy/Leverhulme Small Grant awarded to RMH (SG150283) and a British Academy International Partnerships and Mobility Scheme grant awarded to SLH (PM140174). MT acknowledges support from the National Research Foundation of South Africa. MT is a lead investigator with the Centre of Excellence in Human Development, University of Witwatersrand, South Africa.

**Competing interests** None declared.

**Patient consent** Parental/guardian consent obtained.

**Ethics approval** University of Bath and Stellenbosch University Research Ethics Committees.

**Provenance and peer review** Not commissioned; externally peer reviewed.

**Data sharing statement** No additional data are available.

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
