## [Reviewer comments · BMJ Open]

ARTICLE DETAILS

TITLE (PROVISIONAL)	Post-Trauma Coping in the Context of Significant Adversity: A Qualitative Study of Young People Living in an Urban Township in South Africa
AUTHORS	Hiller, Rachel; Halligan, Sarah; Tomlinson, Mark; Stewart, Jackie; Skeen, Sarah; Christie, Hope

VERSION 1 - REVIEW

REVIEWER	Rebekah Ogilvie Shock Trauma Service Canberra Hospital & Health Services Australian Capital Territory Australia
REVIEW RETURNED	23-Mar-2017

GENERAL COMMENTS	This topic and the use of qualitative methods in trauma research are highly important and I commend you for examining these phenomena in this way, particularly as many readers and researchers alike that are new to qualitative research, struggle to find the 'so what factor'. What you have demonstrated is that indeed the most illuminating qualitative findings go further than description, they interpret, they explain and they solve problems. Your paper is robust and well written and it was a pleasure to review.
---

REVIEWER	Marieke Sleijpen Utrecht University and Foundation Centrum '45, the Netherlands
REVIEW RETURNED	21-May-2017

GENERAL COMMENTS	The authors discuss a relevant topic and they are transparent with regard to different aspects of their method; my compliments. Nevertheless, there is room for improvement. I outlined a number of comments below. * Was redundancy, or saturation of the data, achieved ? * The method section is well written and the researchers clearly describe how the findings emerged from the data. Nevertheless, I miss this accuracy in the results section. For example: how to interpreted the undetermined numbers - what do they mean by many cases, or many participants, or in some cases? Perhaps the exact numbers can be given as well.
---

	* In the results section the authors conclude that social support was central to coping. But it was also a central theme during the interviews. (The interview guide contains questions about social support.) So is it a logical finding created by the design of their study? “Almost all participants discussed the role of someone in their social circle who had provided them support” because the researchers literally asked them to do this? * What do the authors mean by “trust and shared traumatic experiences were key to this peer support” (p. 13)? Can they elucidate this? Did participants trust their peers? Traumatized youth find it often difficult to trust others (e.g. Anstiss & Ziaian, 2010). * Theme two “forgetting” contains some contradiction. How does “talking to someone about the experience” (p. 15) fits in this theme called forgetting? Although the quote of participant L illustrates it, I think it needs more elaboration. Also in the discussion section I miss the link with other studies about avoidance and its "Janus-Face". Besides, in the discussion section the authors wrote that “avoiding talking would ultimately cause the young person more harm in the long term” (p.23), while I did not read/extract this in/from the results section? * The authors claim that there are relatively few studies about young person’s post-trauma support in LMICs, but the authors miss a significant body of research in the area of for example displaced persons or resilience related to LMIC’s or other contexts involving uncertainty and/or danger. The discussion section needs more embedding in the larger picture - other literature that captures the same themes. What does this study add? * As the authors assert: for developing intervention/prevention programs it is important to understand the complexities of the environment faced by young people in LMICs. So what do the findings of this study tell us; what are the practical implications of these findings. And what are (probably) the differences compared to practical implications for HICs relating to post-trauma support?
--	--

REVIEWER	Dr Saskia Keville University of Hertfordshire United Kingdom
REVIEW RETURNED	26-May-2017

GENERAL COMMENTS	This was an interesting study, given the increasing diversity in the UK, understanding trauma within other cultural contexts is of value to understanding how to adapt interventions in the UK in addition to how to support trauma in other contexts. The one area that would strengthen the discussion is greater exploration of the value/hindering aspects of 'forgetting' and how this might differ in HICs and LMICs. Here one could explore the literature on experiential avoidance. An important theme was the potential for a lack of justice and with this perpetrators living in the vicinity, thus impacting on a sense of safety. clearly the only way to manage this was avoidance, which was highlighted as a helpful response by family members. In HIC, the literature talks about the unhelpful aspects of avoidance, yet clearly in this LMIC context it serves a
--

	helpful function and this could be drawn out further, such that cultural issues are understood and addressed in the interventions that we might provide in the UK, eg for refugees, and the interventions westerners might apply in LMIC contexts (which might be helpful in HIC cultures but not in LMIC ones).
--	--

VERSION 1 – AUTHOR RESPONSE

Reviewer: 1

Reviewer Name: Rebekah Ogilvie

Institution and Country: Shock Trauma Service, Canberra Hospital & Health Services, Australian Capital Territory, Australia Please state any competing interests: None declared

Thank you for your submission which seeks to explore post-trauma support and coping of young people living in a high adversity settlement in South Africa. This topic and the use of qualitative methods in trauma research are highly important and I commend you for examining these phenomena in this way, particularly as many readers and researchers alike that are new to qualitative research, struggle to find the 'so what factor'. What you have demonstrated is that indeed the most illuminating qualitative findings go further than description, they interpret, they explain and they solve problems. Your paper is robust and well written and it was a pleasure to review. Thank you for your positive review of this paper. No further changes required.

Reviewer: 2

Reviewer Name: Marieke Sleijpen

Institution and Country: Utrecht University and Foundation Centrum '45, the Netherlands Please state any competing interests: none declared

The authors discuss a relevant topic and they are transparent with regard to different aspects of their method; my compliments. Nevertheless, there is room for improvement. I outlined a number of comments below.

Thank you for your review of this manuscript. Below we outline how each point has been addressed. Changes in the manuscript have been highlighted in red.

Was redundancy, or saturation of the data, achieved ?

Saturation of the data was achieved, which has now been highlighted in the text (page 9).

The method section is well written and the researchers clearly describe how the findings emerged from the data. Nevertheless, I miss this accuracy in the results section.

For example: how to interpreted the undetermined numbers - what do they mean by many cases, or many participants, or in some cases? Perhaps the exact numbers can be given as well. Using thematic analysis and based on good practice in qualitative research we have not included specific numbers who endorsed various themes, as that would be quantifying the qualitative information. In qualitative sampling expressing results in frequencies (i.e., 10 out of 20 participants said X) can be misleading (Pope, Ziebland & Mays, 2000). The aim of this qualitative data is to aid our understanding of participant experiences and processes (Burnard et al., 2008; Marks & Yardley, 2004). That said, we have made some changes in the Themes section to clarify whether the theme

was endorsed by most of the participants or where it might be (for example) specific to only a few participants or to a specific group (e.g., a specific trauma type).

In the results section the authors conclude that social support was central to coping. But it was also a central theme during the interviews. (The interview guide contains questions about social support.) So is it a logical finding created by the design of their study? “Almost all participants discussed the role of someone in their social circle who had provided them support” because the researchers literally asked them to do this?

The participants were not directly asked about who in their social circle provided them with support. Questions were open (page 8) – e.g., “Was there anything you did to try and make yourself feel better or different?” or “Did you talk to anyone about how you were feeling?”. That is, the questions were open to allowing the young person to either discuss social support or say that they received no social support. Follow up questions were also generic – e.g., “Was it helpful or not helpful?”. Thus, that almost all discussed the importance of social support to their coping was considered an important and central theme.

We have slightly edited the title to “Post-Trauma Coping in the Context of Significant Adversity...” removing the word ‘support’ to better reflect the broader purpose of the research.

What do the authors mean by “trust and shared traumatic experiences were key to this peer support” (p. 13)? Can they elucidate this? Did participants trust their peers? Traumatized youth find it often difficult to trust others (e.g. Anstiss & Ziaian, 2010).

This has been clarified on page 13 and in the Discussion (page 23-24).

Theme two “forgetting” contains some contradiction. How does “talking to someone about the experience” (p. 15) fits in this theme called forgetting? Although the quote of participant L illustrates it, I think it needs more elaboration. Also in the discussion section I miss the link with other studies about avoidance and its “Janus-Face”. Besides, in the discussion section the authors wrote that “avoiding talking would ultimately cause the young person more harm in the long term” (p.23), while I did not read/extract this in/from the results section?

The importance of “forgetting” was indeed a complex theme as it could be adaptive or maladaptive (as separated out under Theme 2, page 15), but was considered central to coping. “Talking about the experience” was captured under this code as this is the information that was generated by the participants – i.e., that talking was helpful because it would mean they could then ‘forget’ what happened. We have made some changes on page 15 to clarify this point. We have also replaced the quote by Participant A, with a quote that better reflects this theme.

The Discussion has also been extended to discuss the role of “forgetting” / avoidance.

The authors claim that there are relatively few studies about young person’s post-trauma support in LMICs, but the authors miss a significant body of research in the area of for example displaced persons or resilience related to LMIC’s or other contexts involving uncertainty and/or danger.

The discussion section needs more embedding in the larger picture - other literature that captures the same themes. What does this study add?

The Discussion has been extended to include more comments about other literature on LMIC contexts and high-risk contexts in HICs (page 23).

As the authors assert: for developing intervention/prevention programs it is important to understand the complexities of the environment faced by young people in LMICs. So what do the findings of this study tell us; what are the practical implications of these findings. And what are (probably) the differences compared to practical implications for HICs relating to post-trauma support?

The Discussion has been edited to extend more on these implications (page 23-26).

Thank you for your feedback on this manuscript.

Reviewer: 3

Reviewer Name: Dr Saskia Keville

Institution and Country: University of Hertfordshire, United Kingdom Please state any competing interests: None declared

This was an interesting study, given the increasing diversity in the UK, understanding trauma within other cultural contexts is of value to understanding how to adapt interventions in the UK in addition to how to support trauma in other contexts. The one area that would strengthen the discussion is greater exploration of the value/hindering aspects of 'forgetting' and how this might differ in HICs and LMICs. Here one could explore the literature on experiential avoidance. An important theme was the potential for a lack of justice and with this perpetrators living in the vicinity, thus impacting on a sense of safety. clearly the only way to manage this was avoidance, which was highlighted as a helpful response by family members. In HIC, the literature talks about the unhelpful aspects of avoidance, yet clearly in this LMIC context it serves a helpful function and this could be drawn out further, such that cultural issues are understood and addressed in the interventions that we might provide in the UK, eg for refugees, and the interventions westerners might apply in LMIC contexts (which might be helpful in HIC cultures but not in LMIC ones).

VERSION 2 – REVIEW

REVIEWER	Marieke Sleijpen Foundation Centrum '45, the Netherlands and Utrecht University, the Netherlands
REVIEW RETURNED	06-Jul-2017

GENERAL COMMENTS	It is an interesting and relevant article; it was a pleasure to review.
---

REVIEWER	Dr Saskia Keville University of Hertfordshire, United Kingdom
REVIEW RETURNED	20-Jul-2017

GENERAL COMMENTS	This paper is much clearer following the amendments and provides valuable information on trauma and the differing ways of managing this for services based in LMICs and also HICs with increasing refugee populations.
--